

# From R-squared to coefficient of model accuracy for assessing "goodness-of-fits"

Charles Onyutha[1]

[1]Department of Civil and Building Engineering, Kyambogo University, P.O Box 1, Kyambogo, Kampala, Uganda

*Correspondence to*: (conyutha@kyu.ac.ug)

**Abstract.** Modelers tend to focus more on advancing methods of statistical and mathematical modeling than developing novel techniques for comparing modeled results with observations or establishing metrics for model performance assessment. Perhaps solely the most extensively applied "goodness-of-fit" measure especially for assessing performance of regression models is the coefficient of determination $R^2$. Normally, high $R^2$ tends to be associated with an efficient model.

Nevertheless, $R^2$ has been cited to have no importance in the classical model of regression. Even in its use in descriptive statistics, $R^2$ is known to have questionable justification. $R^2$ is inadequate in assessing model performance because it does not give any information on the model residuals. Furthermore, $R^2$ can be low for an effective model. Contrastingly, a very poor model fit can yield high $R^2$. Regressing $X$ on $Y$ yields $R^2$ which is the same as that if $Y$ is regressed on $X$ thereby invalidating its use as a coefficient of determination. Taking into account the drawbacks of using $R^2$, this paper introduces coefficient of

model accuracy (CMA) the derivation of which comprises an analogy to the $R^2$. However, instead of simply squaring an ordinary Pearson's product-moment correlation coefficient to obtain $R^2$, CMA comprises the product of nonparametric sample correlation and model bias. Acceptability of the introduced method can be found demonstrated through comparison of results from simulations by hydrological models calibrated using CMA and other existing objective functions. MATLAB and R codes as well as an illustrative MS Excel file to compute the CMA were provided.

**1 Introduction**

Despite the advances in methods of statistical and mathematical modeling, Alexander et al. (2015) asserts that there continues to exist considerable lack of focus on improving ways to judge the quality of the models. Here, quality can be thought of in terms of  how well a model fits a set of observations and this can be described as "goodness-of-fit". The coefficient of determination (hereinafter denoted as $R^2$ or interchangeably used with R-squared) is perhaps solely the most

extensively applied measure of "goodness-of-fit" especially for regression models (Kvålseth, 1985). In various studies from the various disciplines of Geosciences (also called Earth sciences) such as atmospheric science, hydrology, and environmental science, R-squared is commonly applied in many studies. For instance in hydrology, some of the studies which applied correlation and/or $R^2$ to evaluate model performance include Chen et al. (2019), Lane et al. (2019), Faiz et al. (2018), Krysanova et al. (2018), Nagraj et al. (2018), Unduche et al. (2018), Bennett et al. (2013), and Ritter and Muñoz-





Carpena (2013). In various studies, researchers also commonly use correlation and/or $R^2$ to evaluate reanalyses or satellite precipitation products. Examples of such studies include Fallah et al. (2020), Shaodan et al. (2020), Irvem and Ozbuldu (2019), Trinh-Tuan et al. (2019), Zandler et al. (2019), Derin et al. (2017), Jiang et al. (2017), Wang et al. (2017), Peña-Arancibia et al. (2013), and Ward et al. (2011).

       In assessments of "goodness-of-fits", there is the common tendency to associate a high value of $R^2$ to an effective

model (Quinino et al., 2013). However, there have been very strong statements made by some researchers such as Goldberg (1991), and Cameron (1993) on the use of $R^2$ to measure "goodness-of-fit". Goldberg (1991) asserted that "the most important aspect of $R^2$ is that it has no importance in the classical model of regression". Furthermore, $R^2$ is not a statistical test, and there seems to be no intuitive justification for its use as a descriptive statistic" (Cameron 1993), suggesting that the value of $R^2$ should not even be reported (Quinino et al., 2013). The use of correlation-based approaches (such as $R^2$) for

evaluating model performance has been argued by a number of researchers such as Legates and Davis (1997) and Willmott (1981) to be an inappropriate practice. Further insights on the limitations of $R^2$ were given by some researchers such as Sherri and McGuire (2019), Li (2017), Alexander et al. (2015), Krause et al. (2005), Schemper (2003), Golbraikh (2002), Weglarczyk (1998), and Cameron and Windmeijer (1997). A few reasons why $R^2$ is inadequate to assess predictive power of models are that: $R^2$ can be low for an accurate model, and on the other hand, an inaccurate model can yield high $R^2$ (Shalizi,

2015). Another problem is that the value of $R^2$ when we regress $X$ on $Y$ turns out to be the same as the $R^2$ attainable by regressing $Y$ on $X$. This means that $R^2$ cannot be taken to be indicative of the variance in observations explained by a model, thereby invalidating the use of $R^2$ as the coefficient of determination. Furthermore, the value of $R^2$ does not give any information on the model residuals. In other words, $R^2$ only quantifies dispersion but not bias in the data. For instance, if we take 10% of observed data to act as modeled results, the $R^2$ value will still be 100% despite the 90% bias. This means that $R^2$

should always be accompanied with residual analyses. The common practice is to use residual plots to give an insight on the bias. Furthermore, support for the analyses of residuals tend to be independently obtained using Root Mean Squared Error (RMSE), and the Model Average Bias (MAB). By the time of writing this paper, there was no formula available which addresses the shortcomings of $R^2$ by comprehensively quantification of dispersion and the measure of bias.

       Therefore, this study aimed at introducing coefficient of model accuracy (CMA) the derivation of which bears an

analogy to the well-known $R^2$. The formula of CMA was carefully derived as a product of non-parametric correlation coefficient and the measure of model bias. CMA varies over the range 0–1. The remaining parts of this paper were organized as follows. Section 2 comprises a stepwise derivation of CMA. Comparison of CMA and other "goodness-of-fit" measures was made using a modeling and simulation case study as presented in Section 3. Results and discussion of the case study were presented in Section 4. Finally, conclusions were made in Section 5.





## 2 Materials and methods

### 2.1 The new approach

#### 2.1.1 Basis of the CMA

Let one variable, for instance the modeled series of sample size $n$, be denoted by $Y$. Again, let the other variable (or the observed series) be denoted by $X$. If $\overline{x}$ and $\overline{y}$ are the mean values of the $x_i$'s and $y_i$'s and $r$ refers to the sample Pearson's product-moment correlation coefficient while $m$ is the least squares linear regression slope, $r$ and $m$ can be given by

$$r = \frac{\sum_{i=1}^{n}\left(x_i - \overline{x}\right)\left(y_i - \overline{y}\right)}{\sqrt{\sum_{i=1}^{n}\left(x_i - \overline{x}\right)^2 \sum_{i=1}^{n}\left(y_i - \overline{y}\right)^2}} \tag{1}$$

$$m = \frac{\sum_{i=1}^{n}\left(x_i - \overline{x}\right)\left(y_i - \overline{y}\right)}{\sum_{i=1}^{n}\left(x_i - \overline{x}\right)^2} \tag{2}$$

Based on a cursory look at Eqs. (1) and (2), $r$ and $m$ are noticeably related via the co-variance of $X$ and $Y$ or $n \times Cov(X,Y) = \sum_{i=1}^{n}\left(x_i - \overline{x}\right)\left(y_i - \overline{y}\right)$. In other words, if we make the numerators the subjects in Eqs. (1) and (2) and equate them,

$$r = m \times \sqrt{\frac{\sum_{i=1}^{n}\left(x_i - \overline{x}\right)^2}{\sum_{i=1}^{n}\left(y_i - \overline{y}\right)^2}} \tag{3}$$

If we let $R^2$ denote the coefficient of determination based on the ordinary (for instance, the Pearson's) correlation coefficient $r$, while $\gamma$ is the ratio of $\sum_{i=1}^{n}\left(x_i - \overline{x}\right)^2$ to $\sum_{i=1}^{n}\left(y_i - \overline{y}\right)^2$, squaring both sides of Eq.(3) yields

$$R^2 = m^2 \times \gamma = m^2 \times \frac{\sum_{i=1}^{n}\left(x_i - \overline{x}\right)^2}{\sum_{i=1}^{n}\left(y_i - \overline{y}\right)^2} \tag{4}$$

Highlights of a few reasons why $R^2$ is not suitable for use as a "goodness-of-fit" can be noticed from Eq. (4). First of all, the deviations of $X$ and $Y$ from their means to obtain $\gamma$ are assessed independently, thus, the model errors are not taken into account. Secondly, possible outliers in $X$ or $Y$ would influence $m$, $\gamma$ and, eventually $R^2$. Analogous to the expression in Eq. (4), this paper proposes a new metric CMA.





### 2.1.2 Step-wise derivation of CMA

The first step is the transformation of modeled data $Y$. To do so, consider $u_i$ as the number of times the $i^{\text{th}}$ data point is exceeded by others within the given sample. Again, for the given sample, take $v_i$ as the number of times the $i^{\text{th}}$ data point exceeds others. Let $e_i$ denote the number of times the $i^{\text{th}}$ data point appears within the sample. We can non-parametrically transform $Y$ and $X$ in terms of the difference between exceedance and non-exceedance counts of data points to obtain series $d_y$ and $d_x$, respectively using

$$d_{y,i} = v_{y,i} - u_{y,i} = n - e_{y,i} - 2u_{y,i} \qquad \text{for } 1 \leq i \leq n \tag{5}$$

$$d_{x,i} = v_{x,i} - u_{x,i} = n - e_{x,i} - 2u_{x,i} \qquad \text{for } 1 \leq i \leq n \tag{6}$$

where $u_y$, $v_y$, and $e_y$ denote $u$, $v$, and $e$ applied to $y$'s. Similarly, the $u_x$, $v_x$, and $e_x$ represent $u$, $v$, and $e$ applied to $X$. For an illustration, consider the dataset $Y$ with $n = 10$ such that $y = \{5, 8, 3, 6, 2, 5, 8, 1, 4, 6\}$. It means $u_y = \{4, 0, 7, 2, 8, 4, 0, 9, 6, 2\}$, $v_y = \{4, 8, 2, 6, 1, 4, 8, 0, 3, 6\}$, $e_y = \{2, 2, 1, 2, 1, 2, 2, 1, 1, 2\}$, and $d_y = \{0, -8, 5, -4, 7, 0, -8, 9, 3, -4\}$.

The second step entails computation of non-parametric linear trend slope $f$ for regression of $Y$ on $X$. The transformations in Eqs. (5) and (6) lead to a special case in which the non-parametric correlation between $X$ and $Y$ is the same as $f$. If we substitute the $x$ and $y$ in Eq. (1) with $d_x$ and $d_y$, respectively, $f$ can be obtained using

$$f = \frac{\sum_{i=1}^{n}(d_{y,i} \times d_{x,i})}{\sqrt{\sum_{i=1}^{n} d_{y,i}^2 \times \sum_{i=1}^{n} d_{x,i}^2}} \qquad \text{for} \quad 1 \leq i \leq n \tag{7}$$

It is worth noting that the $d_y$ and $d_x$ from Eqs. (5) and (6), respectively, can be comparable to simply ranking $Y$ and $X$ in 95  descending order. The difference between simply ranking the data in descending order and applying transformation using Eqs. (5) or (6) is that the ordinary data ranks would all be positive and with their mean not equal to zero; however, $d_y$ or $d_x$ has a mean of zero and varies over the range $\mp(n-1)$. Thus, the $f$ in Eq. (7) is simpler to compute using $d_y$ and $d_x$ than when ordinary ranks of $Y$ and $X$ are considered. The values of $f$ ranges from -1 to 1. When $f = 0$, it means $X$ and $Y$ are totally uncorrelated or the non parametric linear regression slope is zero. The cases $Y = -X$ and $Y = X$ yield $f = -1$ and 100  $f = 1$, respectively. Importantly, unlike $r$ from Eq. (1), $f$ (Eq. 7) is not susceptible to possible outliers in the data.

The third step of deriving CMA includes quantification of the measure of model bias. Ordinarily, model bias or error tends to be in terms of the difference between the modeled and observed data points. In this case, summation of the errors requires normalization by a measure of the variability in the observations. This is the basis for a number of metrics such as MAB, RMSE, and NSE which all present the common issue of not having their values within the "standard" range of zero to one.





For the new method, penalties are first assigned to the modeled series and model errors obtained in terms of the deviations from the mean of observed data. Let $H$ be a new series derived from the modeled data by taking into account some penalty. For every positive observed value, if the modeled data point is negative (or vice versa), a penalty is awarded. Expressly, for $1 \leq i \leq n$,

$$h_i = \begin{cases} 0 & \text{if } \left(x_i < 0 \text{ and } y_i \geq 0\right) \text{ or } \left(x_i > 0 \text{ and } y_i \leq 0\right) \\ y_i, & \text{otherwise} \end{cases} \tag{8}$$

Let the comparison baseline $\xi$ be a function of $\overline{x}$ for instance $\xi = \overline{x}$, $\xi = 2\overline{x}$, and $\xi = 3\overline{x}$. Here, $\overline{x}$ as defined in Eq. (1) is based on the original or untransformed $X$. Consider the deviations $\omega_1$ and $\omega_2$ to be used in expressing the measure of model bias $\beta$. If *min* and *max* denote the minimum and maximum of any two values,

$$\omega_{1,i} = \left(min\left(h_i, x_i\right) - \xi\right)^2 \tag{9}$$

$$\omega_{1,i} = \left(max\left(h_i, x_i\right) - \xi\right)^2 \tag{10}$$

Normally errors of opposite signs can cancel each other during their summation. This was avoided in Eqs. (9)–(10) by squaring of the terms in brackets in the right hand side. Based on the magnitude of the deviation of $h_i$ or $x_i$ from $\xi$ considering $1 \leq i \leq n$, we can obtain $\omega_{1,i} > \omega_{2,i}$, $\omega_{1,i} < \omega_{2,i}$, and $\omega_{1,i} = \omega_{2,i}$. To ensure minimum and maximum deviations are summed up separately, the terms $\theta_1$ and $\theta_2$ were considered such that for $1 \leq i \leq n$,

$$\theta_{1,i} = min\left(\omega_{1,i}, \omega_{2,i}\right) \tag{11}$$

$$\theta_{2,i} = max\left(\omega_{1,i}, \omega_{2,i}\right) \tag{12}$$

therefore, the measure $\beta$ of model bias can be given by

$$\beta = \begin{cases} 0 & \text{if } \sum_{i=1}^{n} h_i = 0 \ or \ \sum_{i=1}^{n} \theta_{2,i} = 0 \\ \left(\sum_{i=1}^{n} \theta_{1,i} \times \left(\sum_{i=1}^{n} \theta_{2,i}\right)^{-1}\right)^2, & \text{otherwise} \end{cases} \tag{13}$$

The values of $\beta$ ranges from zero to one. An ideal model yields $\beta = 1$. When $\beta = 0$, it means the outputs of the model can simply be represented by the mean of observed data. Analogous to Eq. (4), the CMA can be computed using,


$$CMA = f^2 \times \beta = \frac{\left( \sum_{i=1}^{n} \left( d_{y,i} \times d_{x,i} \right) \right)^2}{\sum_{i=1}^{n} d_{y,i}^2 \times \sum_{i=1}^{n} d_{x,i}^2} \times \left( \frac{\sum_{i=1}^{n} \theta_{1,i}}{\sum_{i=1}^{n} \theta_{2,i}} \right)^2$$ (14)

The values of CMA ranges from 0 to 1. CMA equal to one indicates a perfect model (no errors). However, CMA equal to zero indicates that the model is not better than the comparison baseline (such as the mean of observed data). An important note on selection of $\xi$ is that putting $\xi \leq \overline{x}$ in Eqs. (9) and (10) makes $\omega_1$ and $\omega_2$ conservative. However when $\xi \geq 3\overline{x}$, the values of $\omega_1$ and $\omega_2$ become exaggerated. In other words, $\xi \leq \overline{x}$ makes CMA approach its maximum value of one

slowly. On the other hand, with $\xi \geq 3\overline{x}$, CMA approaches 1 fast. Therefore, to ensure that $\omega_1$ and $\omega_2$ are neither conservative nor amplified, $\xi = 2\overline{x}$ was adopted in this paper.

### 3 Case study

#### 3.1 Data and selected models

It is a common practice to compare a new method with existing ones in geosciences. To do so, quality controlled daily
hydro-meteorological data consisting of the Blue Nile flow observed at El Diem, as well as potential evapotranspiration (PET) and rainfall over and around the basin were adopted in a catchment-wide form from a previous study (Onyutha, 2016). The adopted PET, river flow and rainfall were daily series covering the period 1980-2000. Two hydrological models selected were selected to generate "goodness-of-fits" for comparison. These models included the Hydrological Model focusing on Sub-flows' Variation (HMSV) of Onyutha (2019), and Nedbør-Afstrømnings-Model (NAM) (Danish Hydraulic Institute
DHI, 2007; Madsen, 2000; Nielsen and Hansen, 1973). These models were adopted for illustration because of their lumped conceptual frameworks or structure which are compatible with the adopted catchment-wide averaged PET and rainfall. Daily PET and rainfall were used as model inputs. The model output was runoff and this was compared with the observed river flow.

#### 3.2 Comparison of the CMA with other "Goodness-of-fits"

For the calibration of HMSV and NAM, the strategy for automatically changing the model parameters was based on the Generalized Likelihood Uncertainty Estimation (GLUE) of Beven and Binley (2001). As a Bayesian approach, GLUE required several sets of model parameters which were randomized within stipulated limits. For each set of parameters, an objective function (also herein taken as the "goodness-of-fits") was selected and the HMSV or NAM run to obtain 5000 sets of simulations. The objective functions included the CMA (Eq. 14), the ordinary coefficient of determination $R^2$ (Eq. 4),





RMSE (Eq. 15), MAB (Eq. 16), NSE (Nash and Sutcliffe, 1970) Efficiency NSE (Eq. 17), and IoA (Willmott, 1981) IoA (Eq. 18) proposed the  such that

$$RMSE = \sqrt{\frac{1}{n}\sum_{i=1}^{n}\left(x_i - y_i\right)^2} \tag{15}$$

$$MAB = \frac{1}{n}\sum_{i=1}^{n}\left(\frac{y_i - x_i}{x_i}\times 100\right) \tag{16}$$

$$NSE = 1 - \frac{\sum_{i=1}^{n}\left(x_i - y_i\right)^2}{\sum_{i=1}^{n}\left(x_i - \bar{x}\right)^2} \tag{17}$$

$$IoA = 1 - \frac{\sum_{i=1}^{n}\left(x_i - y_i\right)^2}{\sum_{i=1}^{n}\left(\left|x_i - \bar{x}\right| + \left|y_i - \bar{x}\right|\right)^2} \tag{18}$$

The number of parameter sets for which each hydrological model was run was set to 5000. The optimal parameters were those in the set which yielded the best value of the objective function. In other words, the best set of parameters was obtained with the maximum values of CMA and R-squared, while RMSE and MAB were required to at their minimum. The calibration was able to yield 5000 values of each "goodness-of-fit" for comparison.

For further analyses, simulated modeled was obtained based on the calibration using each of the objective functions. Comparison of the "goodness-of-fits" was made in terms of the difference between observed and modeled flow computed using 8 criteria or metrics annual maximum series (MMaxS), annual minimum series (MMinS), long-term mean of the daily series, Coefficient of Variation (CV), skewness, kurtosis, and Inter-Quartile range (IQR).

The "goodness-of-fits" were ranked such that the one with the lowest absolute difference was given a rank of 1. In other words, a rank of 6 was given to the "goodness-of-fit" which yielded the largest absolute difference. The sum of ranks from all the 8 criteria was obtained. The "goodness-of-fit" with the lowest sum of ranks was considered to yield the best modeled results.

## 4 Results and discussion

### 4.1 Comparing CMA with R-squared, RMSE, and MAB

Figure 1 shows plots of each of the "goodness-of-fit" measures (MAB, RMSE, CMA and R-squared) against the other. The relationship between RMSE and MAB is linear with a negative slope (Fig. 1a). This means the values of both RMSE and



MAB are large in magnitude for a poorly fit model. However, with increasing performance of the model, both RMSE and MAB tend toward zero. However, based on whether the model over-estimates or under-estimates the observed data points, MAB can take any positive or negative value, respectively. From Fig.1b, it is evident that for MAB of large magnitude, the

CMA is zero indicating a model with a very poor fit. However, as the magnitude of MAB reduces, CMA increases such that for the set of optimal parameters, MAB and CMA tend o zero and one, respectively. The relationship between RMSE and CMA (Fig. 1c)  was found to follow a power function. Just like for the MAB, CMA is zero for large values of RMSE (Fig. 1c) indicating poor model fit. With improvement in the model performance, RMSE reduces while CMA tends towards 1.

An ideal model yields MAB or RMSE of zero. However, both RMSE and MAB do not have standard ranges. RMSE takes any value from zero to positive infinity. MAB ranges from negative infinity to positive infinity. Thus, when MAB or RMSE is not zero (which is often the case in normal modeling practices), the judgment of the model performance becomes subjective. Sums-of-squares-based error or deviation statistics such as the mean-absolute deviation and RMSE yield values which are counterintuitive and are therefore inappropriate measures of the typical errors (Mielke and Berry, 2001). This

problem stems from the fact that each squared error may not meaningfully be comparable with other squared errors which lead to the set of squared errors; in other words, the triangle inequality may not be satisfied (Mielke and Berry, 2001). Eventually, there are interpretational difficulties or ambiguities inherent in sums-of-squares-based error statistics (such as RMSE) due to their dependence on the average error and the variability within the set of error magnitudes (Wilmott, 2009).

It is noticeable that R-squared remained high whether RMSE or MAB was high or low (Fig. 1e-f). Furthermore, when CMA remained zero, R-squared varied over a wide range with some values close to 1 (Fig. 1d). This was because the R-squared does not quantify model bias as done by CMA. However, as CMA increased from zero towards one, the variability of the R-squared reduced. This showed that with improvements in the model performance, the variability in observed and simulated data become increasingly comparable. Nevertheless, the general poor performance by the R-squared depicted its limitations

(see Introduction Section) in case it is to be used for assessing model performance. capture model bias (an element which basically influences performance of models).

Generally, the results from the HMSV agree with those of NAM with respect to the overall relationships between the "goodness-of-fits". However, because the HMSV and NAM have different structures and parameters, some slight differences between results from the two selected models the in terms of the magnitudes of the "goodness-of-fits" are noticeable. This

difference is expected and cannot be surprising. In fact, even if one model was to be calibrated against observations from two systems (i.e. catchments in this case), there would still be some differences in model results. This is because, the model inputs (such as rainfall and PET) and physiographic characteristics such as catchment area would differ and eventually, the parameter spaces to lead to model optimality would never be the same.





With respect to comparison of $R^2$ and CMA, an important question to answer is: If *X* is regressed on *Y*, is it similar to

regressing *Y* on *X*? Answering this question requires a close look at Eqs. (1), (4), (7) and (13)–(14). Here, $R^2$ (Eq.4) is

considered equal to the *r* (Eq. 1) squared.  In Eq. (1), the multiplication of $\left( x_i - \overline{x} \right)$ and $\left( y_i - \overline{y} \right)$ just like the product of

$\sum_{i=1}^{n}\left( x_i - \overline{x} \right)^2$ and $\sum_{i=1}^{n}\left( y_i - \overline{y} \right)^2$ is commutative thereby ensuring that the value of *r* (and eventually the $R^2$) remains

the same whether we are considering *X versus Y* or *Y versus X*. Similarly, *f* from Eq. (7) remains unchanged when we

consider *X versus Y* or regressing *Y* on *X*. However, because of the differences in the means of *X* and *Y*, the $\omega_1$ (Eq. 9)

obtained by regressing *X* on *Y* is different from that when *Y* is regressed on *X*. Again, the $\omega_2$ (Eq. 10) obtained through

regression of *X* on *Y* is not the same at that for when *Y* is regressed on *X*. Therefore, so long as means of *X* and *Y* are not the

same, $\beta$ (Eq. 13) and eventually CMA (Eq. 14) obtained by regressing *X* on *Y* is always different from that when *Y* is

regressed on *X*.  This explains why CMA is superior to $R^2$.

>> INSERT Figure 1

**4.2 Comparison of CMA with NSE and IoA**

Figure 2 shows values of NSE, IoA, CMA and R-squared obtained during model calibration. Like in Fig. 1e-f, results of both

the HMSV and NAM are comparable with respect to the relationships between the values of the objective functions. For

negative values of  NSE, CMA was zero (Fig. 2a). However, with improvement in the model fit, both NSE and CMA tended

towards 1. It is worth noting that NSE has a wide range of values. In Fig. 2a, c-d, NSE was as low as −60. Actually, NSE can

yield values down to  −∞. NSE is known to be sensitive to bias in model prediction and is normally influenced by the

outliers if present in the series (McCuen et al. 2006). The suitability of NSE has been on the modelers' radar for decades (see,

for instance, Garrick et al.(1978), Martinec and Rango (1989), Legates and McCabe (1999), Krause et al.(2005), Criss and

Winston (2008), Le Moine (2008), Gupta et al. (2009), Ehret and Zehe (2011), Legates and McCabe (1999), and Legates and

McCabe (2013)). As a result there are several variants of NSE based on its modifications to address some of the issues

related to the use of the original version of Nash and Sutcliffe (1970). For instance,  to overcome the oversensitivity of NSE

to peak high values stemming from the influence of squaring the error terms, logarithmic NSE is widely used (Krause et al.

2005). Another improvement was by Legates and McCabe (1999) by considering the ratio of the sum of absolute (instead of

squared) differences between modeled and observed series divided by the sum of absolute (and not the squared) deviations

of the observed data points from their mean. Despite the improvements, the original NSE continues to be more applicable

than its variants, and thus its adoption in this study.

The IoA increased towards its maximum value of one more rapidly than CMA (Fig. 2b-c). When CMA was zero (or for

negative NSE), the values of IoA were noticeably high up to about 0.8 (Fig. 2b-c). In the same line, for negative NSE



(indicating very poor model fit as depicted by zero CMA), the values of R-squared were high or close to 1 conversely
showing good model performance (Fig. 2d-e). From Fig. 2e, it can be seen that for a small IoA, R-squared exhibited large

values and vice versa. In the same line, for MAB of large magnitude, the IoA was still large (Fig. 2f). Unlike the NSE, other

metrics IoA, and $R^2$ range from 0 to 1. The issues on the use of $R^2$ for model assessment were already given in the

Introduction Section. The original version of IoA (Willmott, 1981) which was adopted for comparison purpose has a major

drawback of giving high values (close to 1) even for poorly fit models (Krause et al., 2005). This was the reason why when

CMA was zero, the IoA was as high as 0.6 (Fig. 2b). To address the problems related to the use of IoA, Willmott et al.

(2012) reformulated the IoA such that the refined metric is bounded by −1 and 1 like the correlation coefficient. In a relevant

communication by Legates and McCabe (2013), they remarked that the refinement of the original IoA by extending its

bound over the range −1 to 0 was unnecessary. Other limitations of the refined of the refined IoA can be found elaborately

given by Legates and McCabe (2013).

>> INSERT Figure 2

### 4.3 Comparison of observed and modeled series

Figure 3 shows observed versus modeled series. The optimal parameters obtained based on calibration by the various

objective functions can be seen from Appendix A1. The corresponding values for the sets of optimal parameters based on the

various objective functions were summarized in  Appendix A2. As often done in hydrological modeling, the full time series

is normally divided into two periods. The first and second periods are for calibration and validation, respectively. However,

in this study, the entire or full time series was used for calibration. This was because the focus of this study was more on the

evaluation of "goodness-of-fits" or objective functions than transferability of model parameters from calibration period to

perform simulations over the validation period. The bias in capturing peak high flows varied among the objective functions

(Fig. 3a-f). For instance, the use of MAB for calibration of the HMSV and NAM respectively led to over-estimations and

under-estimations of high peak flow events (Fig. 3e). Possible contrast in the peak flows from the two models was due to the

differences in model structures and also the parameter spaces considered for calibration. Nevertheless, the results from the

models calibrated using the various objective functions adequately captured the variation in observed flow. This indicated

the acceptability of CMA as a "goodness-of-fit" measure.

>>INSERT Figure 3

Figure 4 shows a results of observed and modeled flow in terms of distribution parameter and hydrological extremes. Apart

from the over-estimation based on MAB, variance and long-term mean of daily flow, CV, MMinS, and IQR were well

captured based on calibrations using the various objective functions (Fig. 4a, d-f, h). It was found that the minimum





difference in variance of observed flow and NAM-based modeled series was obtained when calibration was performed using the IoA as the objective function. R-squared was the best in reproducing the IQR when used to calibrate HMSV. However

for NAM, the largest difference between the IQR of observed and that of modeled series was obtained using R-squared. Again, the use of MAB for calibrating NAM (HMSV) reproduced the most (least) biased MMaxS. Over-estimations for skewness and actual excess kurtosis were slight and large, respectively. Usually when both skewness and actual excess kurtosis are zero, it means that the data follows a normal distribution. As realized from Fig. 4b-c, the observed flow was slightly positively skewed (skewness = 1.60) and leptokurtic (kurtosis = 1.79). This showed that the central peak of the

distribution of observed data was higher and sharper while its tails were fatter and longer than the normal distribution. The under-estimations of skewness and kurtosis could be due to failure of the models to capture the large intermittency (or large difference between maximum and minimum values) in the river flow data. In a data scarce region (like where the study area is located), extreme high and low peak flow events (like those falling outside the IQR) tend to be difficult to capture by hydrological models because of the data limitation and poor quality.

Results also show that on one hand, the use R-squared for calibration led to larger over-estimations of observed variance, MMinS, and IQR (Fig. 4a, f, h) by NAM than HMSV. On the other hand, over-estimations of observed variance, MMinS, and IQR were larger for HMSV than those of NAM (Fig. 4a, f, h). Outputs from two models can not be the same because of the difference in model structures. For HMSV and NAM, the minimum difference between MMaxS of observed and modeled series was obtained using R-squared and MAB, respectively. For an ideal (or completely unbiased) model, the

difference between observed and modeled flow is zero. Model bias can be brought about by various factors such as unrealistic assumptions in developing the modeling concept, observation errors of model inputs, etc. When we consider only one model, results from Fig. 4 were comparable. This, again, showed the acceptability of the introduced "goodness-of-fit" metric.

>>INSERT Figure 4


## 5 Conclusions

This study introduced the CMA for assessing model performance. CMA is obtained as the squared product of non-parametric correlation and the measure of bias. CMA ranges from 0 to 1. CMA = 1 indicates an ideal model (no errors). However, CMA = 0 shows that the model is not better than the comparison baseline (such as the mean of observed data).

Unlike R-squared, regressing $X$ on $Y$ yields CMA which is different from that when $Y$ is regressed on $X$. To demonstrate the suitability of the CMA, two hydrological models HMSV and NAM were calibrated using the GLUE strategy (Beven and Binley, 1992) based on a number of objective functions including CMA, $R^2$, RMSE, MAB, NSE, and IoA. Results from the





calibration were compared among the objective functions. Because the CMA combines quantifications of both the dispersion in the data and model bias, it outperformed $R^2$ and was highly competitive with the other "goodness-of-fit" measures.

Therefore, CMA can be considered as an alternative to $R^2$ in evaluation of model performance.

**Conflict of interest**

The author declares no conflict of interest and no competing financial interests.

**Acknowledgment**

The author is grateful to Copernicus Publications for granting the waiver of article processing charges for this paper.

**Author Contribution statement**

The entire work in this paper was based on the effort of the sole author.

**Code/Data availability**

The MATLAB and R or RStudio codes to implement the new method in included in Appendices B and C. Furthermore, an Ms Excel file comprising illustrations of how CMA can be computed in a step-wise way is included as a supplementary

material.

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







**Figure 1:** Comparison of a) MAB and RMSE, b) CMA and MAB, c) CMA and RMSE, d) CMA and R2, e) $R^2$ and RMSE, and f) $R^2$ and MAB


**Figure 2**: Comparison of a) CMA and NSE, b) CMA and IOA, c) NSE and IOA,  and d) $R^2$ and NSE



**Figure 3:** Time series plots of observed flow and modeled series from HMSV and NAM obtained using a) NSE, b) CMA, c) R-squared, d) RMSE, e) MAB, and f) IoA as objective functions for calibration







**Figure 4:** Comparison of modeled and observed flow in terms of a) variance, b) skewness, c) kurtosis, and d) CV, as well as the mean of e) long-term daily flow, f) MMinS, g) MMaxS, and h) IQR.



## Appendix A: **Model parameters and objective functions**

**Table A1**: Optimal parameters of HMSV

| Model parameter | Parameter limits | | Objective function for calibration | | | | | |
|---|---|---|---|---|---|---|---|---|
| | Lower | Upper | NSE | CMA | R-squared | RMSE | MAB | IOA |
| $S_{max}$ | 135 | 145 | 137.63 | 139.81 | 136.67 | 136.88 | 139.91 | 136.95 |
| $a_1$ | 8 | 10 | 8.84 | 9.69 | 8.17 | 8.19 | 9.47 | 9.45 |
| $t_a$ | 380 | 400 | 397 | 394 | 397 | 384 | 395 | 389 |
| $a_2$ | 4 | 7 | 6.74 | 6.97 | 6.73 | 6.70 | 5.66 | 6.24 |
| $t_b$ | 34 | 38 | 38 | 37 | 35 | 34 | 38 | 36 |
| $a_3$ | 6 | 8 | 6.27 | 6.01 | 6.39 | 6.29 | 6.11 | 6.19 |
| $c_3$ | 1 | 4 | 3.35 | 3.80 | 3.41 | 2.90 | 3.64 | 3.17 |
| $b_3$ | 2 | 7 | 3 | 3 | 5 | 3 | 5 | 4 |
| $t_v$ | 1 | 4 | 3 | 2 | 3 | 4 | 2 | 4 |

**Definition of model parameters:**

| | | | | | |
|---|---|---|---|---|---|
| $a_1$ | : | Baseflow parameter | $a_3$ | : | Overland flow parameter 1 |
| $t_1$ | : | Baseflow recession constant (day) | $t_b$ | : | Interflow recession constant (day) |
| $t_u$ | : | Overland flow recession constant 1 (day) | $a_2$ | : | Interflow parameter |
| $t_v$ | : | Overland flow recession constant 2 (day) | $c_3$ | : | Overland flow parameter 2 |
| $S_{max}$ | : | Maximum limit of soil moisture storage deficit (mm) | | | |

**Table A2**: Optimal parameters of NAM

| Model parameter | Parameter limits | | Objective function for calibration | | | | | |
|---|---|---|---|---|---|---|---|---|
| | Lower | Upper | NSE | CMA | R-squared | RMSE | MAB | IOA |
| $U_{max}$ | 0.2 | 0.5 | 0.387 | 0.489 | 0.448 | 0.395 | 0.206 | 0.491 |
| $L_{max}$ | 8 | 15 | 8.740 | 14.092 | 13.593 | 8.143 | 9.537 | 12.714 |
| $CQ_{OF}$ | 0 | 0.2 | 0.009 | 0.008 | 0.016 | 0.011 | 0.008 | 0.010 |
| $CK_{IF}$ | 4 | 15 | 7.969 | 10.748 | 8.407 | 12.695 | 7.441 | 12.456 |
| $CK_1$ | 20 | 35 | 22 | 21 | 25 | 26 | 30 | 24 |
| $CK_2$ | 1 | 6 | 1.590 | 2.139 | 1.533 | 2.482 | 5.683 | 1.423 |
| $T_{OF}$ | 0.01 | 0.2 | 0.051 | 0.154 | 0.088 | 0.017 | 0.018 | 0.198 |
| $T_{IF}$ | 0.01 | 0.2 | 0.127 | 0.036 | 0.055 | 0.116 | 0.144 | 0.067 |
| $T_G$ | 0.01 | 0.2 | 0.091 | 0.059 | 0.118 | 0.041 | 0.120 | 0.072 |
| $CK_{BF}$ | 48500 | 55000 | 54800 | 50800 | 50900 | 50200 | 53500 | 54300 |

**Definition of model parameters:**

| | | | | | |
|---|---|---|---|---|---|
| $U_{max}$ | : | Maximum surface storage (mm) | $L_{max}$ | : | Maximum lower zone storage (mm) |
| $CQ_{OF}$ | : | Overland flow runoff coefficient | $CK_{IF}$ | : | Time constant for interflow (day) |
| $CK_{BF}$ | : | Time constant for baseflow (day) | $T_{IF}$ | : | Threshold value for interflow |
| $T_{OF}$ | : | Threshold value for overland flow | $T_G$ | : | Threshold value for groundwater recharge |





| $CK_1$ , $CK_2$ : | Time constants for routing overland and interflow using two serially connected linear reservoirs (day) |
| --- | --- |

**Table A3**: Values of objective functions for optimal parameters of HMSV

| Objective | Value of objective function | | | | | |
| --- | --- | --- | --- | --- | --- | --- |
| function | NSE | CMA | R-squared | IOA | RMSE | MAB |
| NSE | 0.857 | 0.522 | 0.891 | 0.960 | 616.551 | 1.011 |
| CMA | 0.840 | 0.566 | 0.804 | 0.952 | 649.625 | 0.928 |
| R-squared | 0.795 | 0.484 | 0.959 | 0.930 | 733.624 | 0.792 |
| RMSE | 0.854 | 0.515 | 0.878 | 0.960 | 612.771 | 0.975 |
| MAB | 0.404 | 0.502 | 0.790 | 0.900 | 1252.726 | 1.484 |
| IoA | 0.844 | 0.491 | 0.876 | 0.959 | 641.590 | 1.097 |

In the row for NSE, it means that NSE was used as objective function during calibration; however, the values of the goodness-of-fits were computed for the optimal parameters.

**Table A4**: Values of objective functions for optimal parameters of NAM

| Objective | Value of objective function | | | | | |
| --- | --- | --- | --- | --- | --- | --- |
| function | NSE | CMA | R-squared | IOA | RMSE | MAB |
| NSE | 0.897 | 0.529 | 0.572 | 0.973 | 519.482 | 1.028 |
| CMA | 0.896 | 0.578 | 0.579 | 0.971 | 522.478 | 0.959 |
| R-squared | 0.649 | 0.506 | 0.875 | 0.910 | 1203.688 | 1.553 |
| RMSE | 0.896 | 0.495 | 0.566 | 0.972 | 522.832 | 1.044 |
| MAB | 0.792 | 0.455 | 0.503 | 0.926 | 740.257 | 0.804 |
| IoA | 0.898 | 0.501 | 0.576 | 0.973 | 518.134 | 1.037 |






**Appendix B**: **MATLAB code to compute CMA**

```matlab
function[cm]=CMA(x,y)
 %x=Observed;
 %y=Modeled;
 %cm: Value of the metric CMA
 %How to call the function:
 %x=rand(100,1);
 %y=x*rand();
 %[cm]=CMA (x,y);

 n=length(x); %sample size of the observed data

 %=============================================
 %INITIALIZATION
 %*********************************************
 Dx=zeros(n,1); %for transformed observed series
 Dy=zeros(n,1); %for transformed modeled series
 ux=zeros(n,1); %number of exceedance - observed series
 uy=zeros(n,1); %number of exceedance - modeled series
 wx=zeros(n,1); %number of times a data point appears in observed series
 wy=zeros(n,1); %number of times a data point appears in modeled series
 h=zeros(n,1);  %Penalty-based series derived from modeled data
 A=zeros(n,1);
 B=zeros(n,1);
 C=zeros(n,1);
 D=zeros(n,1);

 %COMPUTING THE PARAMETRIC COMPONENT OF THE MODEL EFFICIENCY
 %***********************************************************
 x_mean=mean(x); % Mean of observed series

 %Penalty-based series derived from y or modeled series
 %***********************************************************
 for i=1:n
     if (x(i,1)<0 && y(i,1)>=0) || (x(i,1)>0 && y(i,1)<=0)
         h(i,1)=0;
     else
         h(i,1)=y(i,1);
     end
 end
 %===========================================================

 for i=1:n
         A(i,1)= (min(x(i,1),h(i,1))-2*x_mean)^2;
         B(i,1)= (max(x(i,1),h(i,1))-2*x_mean)^2;
         C(i,1)= min (A(i,1),B(i,1));
         D(i,1)= max (A(i,1),B(i,1));
 end

 if (sum(D)==0 || sum(h)==0);
     beta1=0;
 else
     beta1=(sum(C)/sum(D));
 end
```





```
    %TRANSFORMATION OF OBSERVED SERIES
%********************************
    for i=1:n
        wx(i)=sum(x==x(i));  %counting how many times a data point x(i) appears in observed
    series X
        ux(i)=sum(x>x(i));   %counting how many times a given value x(i) is exceeded from an
array X
        Dx(i)=n-wx(i)-2*ux(i);  %this is the transformed observed series
    end

    %TRANSFORMATION OF MODELED SERIES
%********************************
    for i=1:n
        wy(i)=sum(y==y(i));  %counting how many times a data point y(i) appears in from an array
    y
        uy(i)=sum(y>y(i));   %counting how many times a given value y(i) is exceeded from an
array y
        Dy(i)=n-wy(i)-2*uy(i);  %this is the transformed modeled series
    end

    %Computing non-parametric trend slope and CMA
%*******************************************
    if (sum(Dx.^2)==0 || sum(Dy.^2)==0);
        rc=0;
    else
        rc=sum(Dx.*Dy)/sqrt(sum(Dx.^2)*sum(Dy.^2));
end

    cm =(rc)^2*beta1^2;

    end
```





**Appendix C**: **R or RStudio code to compute CMA**

```r
MCA<-function(x,y)
{
#   x<-Observed;
# y<-Modeled;
# cm: Value of the metric CMA
# How to call the function:
# x<-runif(100)
# y<-x*runif(100)
# cm<-MCA(x,y)
# cm

  n<-length(x) # %sample size of the observed data

  #creating empty variables to hold computed values
  A <- matrix(nrow=n, ncol=1)
  B <- matrix(nrow=n, ncol=1)
  C <- matrix(nrow=n, ncol=1)
  D <- matrix(nrow=n, ncol=1)
  wx <- matrix(nrow=n, ncol=1)
  ux <- matrix(nrow=n, ncol=1)
  Dx <- matrix(nrow=n, ncol=1)
  wy <- matrix(nrow=n, ncol=1)
  uy <- matrix(nrow=n, ncol=1)
  Dy <- matrix(nrow=n, ncol=1)
  tDx <- matrix(nrow=n, ncol=1)
  tDy <- matrix(nrow=n, ncol=1)
  tDxy <- matrix(nrow=n, ncol=1)
  h <- matrix(nrow=n, ncol=1)

  # COMPUTING THE PARAMETRIC COMPONENT OF THE MODEL EFFICIENCY
  # ***********************************************************
     x_mean<-mean(x) #% Mean of observed series

   # Penalty-based series derived from y or modeled series
    for (i in 1:n)
    {
      if (x[i]<0 & y[i]>=0)
      { h[i]=0 }
      else if (x[i]>0 & y[i]<=0)
      {h[i]=0}
      else
      {h[i]=y[i]}
    }
    for (i in 1:n)
    {
       A[i]= (min(x[i],h[i])-2*x_mean)^2;
       B[i]= (max(x[i],h[i])-2*x_mean)^2;
       C[i]= min (A[i],B[i]);
       D[i]= max (A[i],B[i])
    }

    if (sum(D)==0 | sum(h)==0)
      { beta1=0 }
    else
    { beta1=(sum(C)/sum(D)) }
```





```
        #TRANSFORMATION OF OBSERVED SERIES
for (i in 1:n)
        {
            wx[i]=sum(x==x[i])   #counting how many times a data point x(i) appears in observed series X
            ux[i]=sum(x>x[i])    #counting how many times a given value x(i) is exceeded from an array X
            Dx[i]=n-wx[i]-2*ux[i] #this is the transformed observed series
}

        #TRANSFORMATION OF MODELED SERIES
        for (i in 1:n)
        {
wy[i]=sum(y==y[i]) #counting how many times a data point y(i) appears in from an array y
          uy[i]=sum(y>y[i])   #counting how many times a given value y(i) is exceeded from an array y
          Dy[i]=n-wy[i]-2*uy[i] #this is the transformed modeled series
        }

for (i in 1:n)
        {
          tDx[i]=(Dx[i])^2
          tDy[i]=(Dy[i])^2
          tDxy[i]=Dx[i]*Dy[i]
}

        #Computing non-parametric trend slope and CMA
        if (sum(tDx==0) | sum(tDy==0))
        { rc=0 }
else
        { rc=sum(tDxy)/sqrt(sum(tDx)*sum(tDy)) }

        cm =(rc)^2*beta1^2

return(cm)
    }
```