# Peer review of "From R-squared to coefficient of model accuracy for assessing "goodness-of-fits""

_Geoscientific Model Development, 2020_

## Short Comment (SC1) · 25 Jun 2020

I am not an expert in statistical measures for "goodness of fits", but I found the paper by C. Onyutha interesting and useful. Therefore, I provide this comment.

I coded the Coefficient of Model Agreement (CMA), introduced in this paper, and compared the outcome to other measures, including Pearson's correlation coefficient ($R^2$) and the Index of Agreement (IoA), both as described in this paper. I also evaluated the often-used Taylor skill score S, see Taylor (2001), Eq. (4). I applied the CMA together with the other measures for an application on which I am working presently (contrail cirrus modelling, in extension of work described earlier (Schumann 2012; Schumann and Graf 2013; Schumann and Heymsfield 2017), further details could be made available

on request). Here I report some related experiences, just for discussion.

My goal is to assess the "goodness" of combined cirrus-contrail model images approximating satellite observation images of cirrus optical thickness and top-of-atmosphere radiances. The simulated and observed contrail features are mostly small-scale structures compared to horizontal cirrus scales, and the model results are sensitive to small wind errors causing contrail displacements at scales comparable to contrail widths. The observations show cirrus properties with both systematic and random deviations from the cirrus model, e.g., because of small-scale processes difficult to treat accurately in any cirrus model (such as turbulence, humidity variability, ice particle habit variability, etc.)

In agreement with the results shown in Figure 2 of this paper, I found low values of CMA for cases with high values of IoA, R2 and S. So, it seems that high values of CMA are obtained only when the model is nearly perfect in representing the observations. I think, it is difficult to achieve high values of CMA when the observations contain random errors, and when the model-observation agreement "goodness" is sensitive to small shifts in small-scale structures. So I found it demanding to find an optimum criterion for "goodness" in this application.

Some technical remarks:

The method uses a count of how many times a data point $x(i)$ appears in observed series $x$, see line 497. In praxis, data may occur with small round-off errors so that $x$ is nearly equal to a set of values in the observations. How can one account for such a near-equality?

I suggest that the author also compares to the Taylor skill score, as given in Eq. (4) of Taylor (2001).

Finally, I ask the author to make sure that all abbreviations used in the text are defined (I missed: IoA).

[Figure]

**References**

Schumann, U., 2012: A contrail cirrus prediction model. Geosci. Model Dev., 5, 543–580, doi: 10.5194/gmd-5-543-2012.

Schumann, U., and K. Graf, 2013: Aviation-induced cirrus and radiation changes at diurnal timescales. J. Geophys. Res., 118, 2404-2421, doi: 10.1002/jgrd.50184.

Schumann, U., and A. Heymsfield, 2017: On the lifecycle of individual contrails and contrail cirrus. Meteor. Monogr., 58, 3.1-3.24, doi: 10.1175/AMSMONOGRAPHS-D-16-0005.1.

Taylor, K. E., 2001: Summarizing multiple aspects of model performance in a single diagram J. Geophys. Res., 106, 7183-7192, doi: 10.1029/2000JD900719.

---

## Author Comment (AC1) · 5 Jul 2020

GENERAL

The author is grateful to Ulrich Schumann for acknowledging that the paper is interesting. The author deemed the comments generated by Schumann constructive for improving the coefficient of model accuracy (CMA) being introduced. Four comments that required response are below.

COMMENT 1

While working on contrail cirrus modeling to extend his previous research studies (Schumann 2012; Schumann and Graf 2013; Schumann and Heymsfield 2017), Schumann noted that high values of coefficient of model accuracy (CMA) are only obtainable

when the model is nearly perfect in representing the observations. He thinks achieving high values of CMA becomes a difficult task when observations contain random errors or in the case where model-observation agreement "goodness" is sensitive to small shifts in small-scale structures. Schumann remarked that it was demanding for him to find an optimum criterion for "goodness" while applying CMA.

REPLY

The author agrees with the remarks from Ulrich Schumann that high values of the version of CMA presented in the discussion paper can be obtained when the model is nearly effective. This may make it demanding to fulfill the criterion set to determine acceptability of model results.

This problem was solved as follows. CMA is now expressed as a function of (i) non-parametric normalized linear regression coefficient (trend slope), (ii) difference between the variances of observed and modeled data, and (iii) difference between the means of observed and modeled data.

In determining the measures of co-variation, the comparison baseline was changed from 2*Ax to 3*Ax where Ax is the mean of the observed data. Stepwise derivation of CMA was revised and can be found in Eqs. (A1)-(A10) (see Figures 1-2) of this document. In the revised manuscript, Eqs. (5)-(14) of the discussion paper will be replaced with Eqs. (A1)-(A10) (see Figures 1 and 2 of this reply). The MATLAB and R codes for computing MCA were revised and can be found provided as supplementary materials to this reply. Furthermore, the MATLAB and R codes in the discussion paper will be revised accordingly during the revision of the manuscript.

Conclusively, it was found that high values of CMA can easily be attained with the revised formula.

COMMENT 2

Schumann suggested that comparisons made in this paper should comprise the Taylor

skill score, as given in Eq. (4) of Taylor (2001).

REPLY

Taylor skill score (TSS) as given in Eq. (4) of Taylor (2001) was included in the comparison of "goodness-of-fits". A part from TSS and CMA, other "goodness-of-fits" included the Nash Sutcliffe Efficiency NSE (Nash and Sutcliffe, 1970), Index of Agreement IOA (Willmott, 1981), Root Mean Squared Error (RMSE), and Model Average Bias (MAB). NSE, RMSE, MAB and IOA can be found in Eqs. (15) to (18) of the discussion paper.

Two rainfall-runoff models including NAM and the Hydrological Model focusing on Sub-flows' Variation (HMSV) of Onyutha (2019) were calibrated using a number of objective functions based on the Generalized Likelihood Uncertainty Estimation GLUE (Beven and Binley, 2001). Apart from CMA (Eq. A9), other objective functions included the Nash Sutcliffe Efficiency NSE (Nash and Sutcliffe, 1970), Index of Agreement IOA (Willmott, 1981), Root Mean Squared Error (RMSE), Model Average Bias (MAB), and TSS (Eq. A11). NSE, RMSE, MAB and IOA can be found in Eqs. (15) to (18) of the discussion paper.

For a selected objective function, each hydrological model was calibrated 5000 times using GLUE strategy. The values of the objective functions were graphically compared. The best modeled series were those obtained using the set of parameters which yielded the best value of the objective function. The best set of parameters was obtained with the maximum values of CMA, TSS, R-squared, and IOA while RMSE and MAB were required to be at their minimum.

Revised results of the "goodness-of-fits" can be found in Figures 3-6 in this document. The detail description of Figures 3-6 will also be included in the revised manuscript. Changes were also made to Tables A1-A4 in the discussion paper but will be included in the revised manuscript.

Specifically, it was found that for a given CMA, values of TSS were generally larger. In

other words, TSS gets closer the maximum value of 1 faster than the CMA.

COMMENT 3

Schumann noted that CMA makes use of the number of times a data point x(i) appears in observed series x, as seen in line 497 of the discussion paper. He remarked that occurrences of data in reality may be characterized by small round-off errors so that x is nearly equal to a set of values in the observations. The question to answer was on how the situation of such a near-equality can be accounted for.

REPLY

Generally, to minimize the effects of round-off errors, the floating point precision can be increased from float to double though this could require large computational resource.

Direct application of arithmetic operators (such as subtraction and addition) to values which are nearly equal can lead to huge round-off errors. This can be a problem for any "goodness-of-fits" not only the CMA. The computer codes provided are purposefully basic to depict the stepwise procedure adopted in the derivation of CMA. However, these codes can provide starting points regarding improvement of the CMA computation procedure to take into account any possible round-off errors (if any). Such an improvement could be in a way to enhance precision without the requirement of large computational resources. Furthermore, dealing with technicality required to minimize round-off errors in computation is deemed to be an open problem in scientific computing. This requires answering questions regarding accuracy, efficiency, precision, robustness, suitable algorithms or computer programs, data structures, and computing architectures.

Instead of using e(i) which represents how many times a data point x(i) appears in observed series x, we can use v(i)-u(i) as expressed in Eqs. (A6)-(A7) (see Figure 2 of this document) MATLAB or R codes included as supplementary materials to this reply.

COMMENT 4

The last comment was that all abbreviations used in the text should be defined.

REPLY

The author recognizes that it was by mistake that abbreviations such as IOA were not defined in the discussion paper by mistake. During revision of the paper, all abbreviations will be defined.

REFERENCES

Beven, K. J., and Binley, A. M.: The future role of distributed models: Model calibration and predictive uncertainty, Hydrol. Process., 6, 279–298, doi:10.1002/hyp.3360060305, 1992.

DHI.: Reference manual, DHI water & environment, in: MIKE11 – A modeling system for rivers and channels, Hørsholm, Denmark: DHI water & environment, pp. 278–325, 2007.

Madsen, H.: Automatic calibration of a conceptual rainfall–runoff model using multiple objectives, J. Hydrol., 235(3–4), 276–288, doi:10.1016/S0022-1694(00)00279-1, 2000.

Nash, J. E., and Sutcliffe, J. V. River flow forecasting through conceptual models part I—a discussion of principles, J. Hydrol., 10, 282–290, doi:10.1016/0022-1694(70)90255-6, 1970.

Nielsen, S. A. and Hansen, E.: Numerical simulation of the rainfall-runoff process on a daily basis, Nordic Hydrol., 4(3), 171–190, 1973.

Onyutha, C.: Hydrological model supported by a step-wise calibration against subflows and validation of extreme flow events. Water, 11, 244, doi:10.3390/w11020244, 2019.

Schumann, U.: A contrail cirrus prediction model. Geosci. Model Dev., 5, 543−580, doi: 10.5194/gmd-5-543-2012, 2012.

Schumann, U., and Graf, K.: Aviation-induced cirrus and radiation changes at diurnal timescales. J. Geophys. Res., 118, 2404−2421, doi: 10.1002/jgrd.50184, 2013.

Schumann, U., and Heymsfield, A.: On the lifecycle of individual contrails and contrail cirrus. Meteor. Monogr., 58, 3.1−3.24, doi: 10.1175/AMSMONOGRAPHS-D-16-0005.1, 2017.

Taylor, K. E.: Summarizing multiple aspects of model performance in a single diagram. J. Geophys. Res., 106, 7183−7192, doi: 10.1029/2000JD900719, 2001.

Willmott, C. J., Robeson, S. M. and Matsuura, K.: A refined index of model performance, Int. J. Climatol., 32, 2088–2094, doi: 10.1002/joc.2419, 2012.

Please also note the supplement to this comment:
https://gmd.copernicus.org/preprints/gmd-2020-51/gmd-2020-51-AC1-supplement.zip

Consider $X$ and $Y$ as the observed and modeled data sets, respectively. Let $H$ be the series resulting from application of penalties to $Y$. Let us take $\overline{x}$ and $\overline{h}$ as the means of the $x_i$'s and $h_i$'s, respectively. The values of $h$ can be derived based on Eq. (8) of the discussion paper.

Adjustment was made to the comparison baseline $\xi$ used in Eqs. (8) and (9) of the discussion paper. Although CMA comprised $\xi = 2\overline{x}$ in the earlier version of CMA, $\xi$ was revised to become $3\overline{x}$. Setting $\xi = 3\overline{x}$ makes CMA to reach its maximum value of 1 faster than when $\xi = 2\overline{x}$ is used. Thus, for $1 \leq i \leq n$, $\omega_{1,i} = \omega_{2,i} = 0$ if $h_i = x_i = \xi$, otherwise

$$\omega_{1,i} = min\left\{\left(h_i - \xi\right)^2, \left(x_i - \xi\right)^2\right\} \tag{A1}$$

$$\omega_{2,i} = max\left\{\left(h_i - \xi\right)^2, \left(x_i - \xi\right)^2\right\} \tag{A2}$$

where $min$ and $max$ respectively denote the minimum and maximum of any two values. Normally errors of opposite signs can cancel each other during their summation. This was avoided in Eqs. (A1)–(A2) by squaring of the terms in brackets on the right hand side. The measure $\beta$ of deviations of the data points from $\xi$ can be given by

$$\beta = \begin{cases} 0 & \text{if } \sum_{i=1}^{n} h_i = 0 \ \ or \ \ \sum_{i=1}^{n} \omega_{2,i} = 0 \\ \sum_{i=1}^{n} \omega_{1,i} \times \left(\sum_{i=1}^{n} \omega_{2,i}\right)^{-1}, & \text{otherwise} \end{cases} \tag{A3}$$

The values of $\beta$ range from zero to one. An ideal model yields $\beta = 1$. When $\beta = 0$, it means the outputs of the model can simply be represented by the mean of observed data. The term $\beta$ gives insight about the co-variation of $X$ and $Y$. Apart from quantifying deviations of the data points from the comparison baseline in terms of $\beta$, we could obtain another expression ($\lambda$) as a function of the standard deviations of $X$ and $Y$ denoted by $\sigma_x$ and $\sigma_y$, respectively such that

$$\lambda = \begin{cases} 0 & \text{if } \sigma_x = \sigma_y = 0 \\ min\left(\sigma_x, \sigma_y\right) \times \left(max\left(\sigma_x, \sigma_y\right)\right)^{-1}, & \text{otherwise} \end{cases} \tag{A4}$$

The values of $\lambda$ vary from zero to one. For the case when the variances of both $X$ and $Y$ are the same, we get $\lambda = 1$. However, when $\lambda = 0$ it means either $\sigma_x = 0$ and $\sigma_y \neq 0$ or $\sigma_x \neq 0$ and $\sigma_y = 0$.

We need another term $\theta$ as a function of $\overline{x}$ and $\overline{h}$ such that

$$\theta = \begin{cases} 1 & \text{if } \overline{h} = \overline{x} = 0 \\ min\left\{\overline{h}^2, \overline{x}^2\right\} \times \left(max\left\{\overline{h}^2, \overline{x}^2\right\}\right)^{-1}, & \text{otherwise} \end{cases} \tag{A5}$$

The term $\theta$ varies from zero to one and takes care of the differences in the means of $X$ and $Y$. When, the means of the observed and modeled series are the same, $\theta = 1$. However $\theta = 0$ means either $\overline{x} = 0$ and $\overline{h} \neq 0$ or $\overline{x} \neq 0$ and $\overline{h} = 0$. The larger the difference between $\overline{x}$ and $\overline{h}$, the smaller the $\theta$ and *vice versa*.

**Fig. 1.** Part 1 of CMA revised derivation

The next step is to compute the linear regression coefficient (or trend slope) such that it varies from zero to one. As already presented in the discussion paper, consider $u_i$ as the number of times the $i^{\text{th}}$ data point is exceeded by others within the given sample. Again, for the given sample, take $v_i$ as the number of times the $i^{\text{th}}$ data point exceeds others. We can non-parametrically rescale $Y$ and $X$ in terms of the difference between exceedance and non-exceedance counts of data points to obtain series $d_y$ and $d_x$, respectively using

$$d_{y,i} = v_{y,i} - u_{y,i} \text{ for } 1 \leq i \leq n \qquad (A6)$$

$$d_{x,i} = v_{x,i} - u_{x,i} \text{ for } 1 \leq i \leq n \qquad (A7)$$

where $u_y$ and $v_y$ respectively denote $u_i$ and $v$ applied to $y$'s. Similarly, $u_x$, and $v_x$ respectively represent $u_i$ and $v$ applied to $X$. For an illustration, consider the dataset $Y$ with $n = 10$ such that $y = \{5, 8, 3, 6, 2, 5, 8, 1, 4, 6\}$. It means $u_y = \{4, 0, 7$ $2, 8, 4, 0, 9, 6, 2\}$, $v_y = \{4, 8, 2, 6, 1, 4, 8, 0, 3, 6\}$, and $d_y = v_y - u_y = \{0, 8, -5, 4, -7, 0, 8, -9, -3, 4\}$.

Non-parametric normalized linear trend slope $f$ for regression of $Y$ on $X$ can be computed using

$$f = \left( \sum_{i=1}^{n} \left( d_{y,i} \times d_{x,i} \right) \right) \times \left( \sqrt{ \sum_{i=1}^{n} d_{y,i}^2 \times \sum_{i=1}^{n} d_{x,i}^2 } \right)^{-1} \qquad (A8)$$

Finally, CMA can be computed using

$$CMA = \left| f \right| \times \beta \times \theta \qquad (A9)$$

where $\left| f \right|$ denotes the absolute value of $f$. The values of CMA range from 0 to 1. CMA equal to one indicates a perfect model (no errors). However, CMA equal to zero indicates that the model is not better than the comparison baseline (such as the mean of observed data).

Instead of using $\beta$, we can use $\lambda$ to computed the alternative coefficient of model accuracy (ACMA) such that

$$ACMA = \left| f \right| \times \lambda \times \theta \qquad (A10)$$

Like CMA, the values of ACMA also range from 0 to 1. For an ideal model (where there are no errors), ACMA is equal to one. When ACMA is zero, it means the modeled series is the same as the mean of the observed data. For data with large variance or coefficient of variation, a good skill score SC can be given by $SC = \beta \times \theta$.

**Fig. 2.** Part 2 of CMA revised derivation

[Figure]

**Fig. 3.** Figure 3: Observed versus modeled flow from (a-g) HMSV and (h-n) NAM rainfall-runoff models calibrated using (a, h) NSE, (b, i) R2, (c, j) CMA, (d, k) IOA, (e, l) RMSE, (f, m) MAB, and (g, n) TSS

[Figure]

**Fig. 4.** Figure 4: Comparison of modeled and observed flow in terms of a) variance, b) skewness, c) kurtosis, and d) coefficient of variation (CV), as well as the mean of e) long-term daily flow, f) minimum fl

**Fig. 5.** Observed versus modeled flow from (a-g) HMSV and (h-n) NAM rainfall-runoff models calibrated using (a, h) NSE, (b, i) R2, (c, j) CMA, (d, k) IOA, (e, l) RMSE, (f, m) MAB, and (g, n) TSS

[Figure]

**Fig. 6.** Comparison of modeled and observed flow in terms of a) variance, b) skewness, c) kurtosis, and d) coefficient of variation (CV), as well as the mean of e) long-term daily flow, f) minimum fl

---

## Referee Comment (RC1) · Anonymous Referee #1 · 7 Sep 2020

Review of: From R-squared to coefficient of model accuracy for assessing "goodness-of-fits" by Charles Onyutha

I have examined part of this paper and found some of it to be well written and interesting. I learned a few things from it, which in my view puts the contribution in the above average category. Nevertheless, I find that it is too long for what it is, and does not make fair comparisons with other metrics. I offer several suggestions as to how it could be improved.

Abstract: the abstract is ineffective as it does not compactly present the major findings. Much of this material reads like an "introduction", starting with the subjective first sentence. My suggested rewrite of this sentence is:

[Figure]

A new measure of "goodness of fit" eliminates several of the well known shortcomings of the widely used correlation coefficient $R^2$, including its insensitivity to bias when models are compared to measurements.

The abstract should then go on to describe the new metric, list its specific properties, then list its similarities, differences, advantages and disadvantages as compared to other metrics. Each sentence should be compact and deliver new and interesting information. No fluff or opinion is appropriate.

Line 45 ff This PP is very interesting, though I am concerned that misinterpretations may be included. Fundamentally, the reason that the values of R or $R^2$ do not depend on whether y is plotted against x, or vice versa, is evident in the definition provided by eq. 1; specifically, there is no difference in how x and y are treated in that mathematical definition, so that x and y are interchangeable and mathematically symmetric. That is, if all of the x's were replaced by y's, and all the y's were replaced by x's, the equation would look the same. Thus, given any two column table of data, either column could be defined as x and the other as y, and the result returned by eq 1 would be the same.

Equation 4 and line 75ff. I have confirmed that Eq 4 is correct, but the claim that the "deviations of X and Y from their means to obtain $\gamma$ are assessed independently" is not. Specifically, the author has mistakenly concealed that dependence in his substitution for "m". The reader can refer back to eq 2 to see that m depends on the PRODUCT of the x and y deviations, and eq 4 depends on m; this conclusively refutes the author's statement.

I don't have time to plough through the author's derivation, but it is clear that computing this would be very difficult compared to many of the simple, single-formula metrics that are currently available to compare models and measurements.

I am unimpressed with the choice of the dataset used to illustrate application of the model. Data that are widely and readily accessible would be better, for example, data from a government website would be better. That original data set could be ma-

nipulated in various simple ways to compare the manipulation, now representing the "model", with the real data, and different metrics applied and compared.

The extended narrative of the performance of different metrics is ineffective. A table whose first column lists the various properties, with additional columns to the right providing values for the particular metric that heads that column, would be far better and would allow direct and simple comparisons of the properties of each metric with all others. Is the metric constrained to range from 0 to 1? Are effects of bias easily excluded? How many different formulae are needed to compute the metric? Ease of computation: if not a trivial calculation, are automated programs readily and widely available, as they are for R^2? Does the metric have real physical significance? Is it widely used? Etc.

My conclusion is that the author has more work to do. A shorter, compact paper would be more effective.

---

## Author Comment (AC2) · 27 Sep 2020

**GENERAL**

The author is thankful to the anonymous reviewer for acknowledging that the contribution of this paper falls in the above average category. The author is further grateful to the reviewer for recognizing that some part of the paper is interesting and well written.

The reviewer remarked that paper is too long for what it is, and does not make fair comparisons with other metrics. To address this comment the reviewer offered a number of suggestions.

**COMMENT 1**

[Figure]

Abstract: the abstract is ineffective as it does not compactly present the major findings. Much of this material reads like an "introduction", starting with the subjective first sentence. My suggested rewrite of this sentence is:

A new measure of "goodness of fit" eliminates several of the well known shortcomings of the widely used correlation coefficient R2, including its insensitivity to bias when models are compared to measurements.

The abstract should then go on to describe the new metric, list its specific properties, then list its similarities, differences, advantages and disadvantages as compared to other metrics. Each sentence should be compact and deliver new and interesting information. No fluff or opinion is appropriate.

REPLY TO COMMENT 1

The author agrees with the reviewer on the need to revise the abstract. In this line, the abstract will be revised as below.

A new measure of "goodness of fit" hereinafter referred to as coefficient of model accuracy (CMA) eliminates several of the well known shortcomings of the widely used coefficient of determination R2, including its insensitivity to bias when model outputs are compared to measurements. CMA can be computed as the product of correlation coefficient, measure of deviations of data points from comparison baseline, and ratio of squared means of observed data (X) to the average of modeled data (Y) or vice versa. Correlation coefficient quantifies the measure of statistical linear relationship between X and Y. Division of squared means of datasets measures the amount by which Y is biased. Measure of deviation of data points from a stipulated comparison baseline gives an insight on the differences in variances of X and Y. This is done while ensuring that CMA yields different values in the two cases (i) when we regress Y on X and (ii) if X is regressed on Y. CMA values ranges from zero to one. Based on large number of simulations, other metrics such as Index of Agreement, and Taylor Skill Score were found to get closer the maximum value of 1 faster than CMA. Comparison of CMA and

other existing metrics can be found given with respect to several properties such as sensitivity to possible outliers, error quantification, and ease of computation. To allow applications of CMA, MATLAB and R codes as well as an illustrative MS Excel file to compute the CMA were provided for readers.

COMMENT 2

Line 45 ff This PP is very interesting; though I am concerned that misinterpretations may be included. Fundamentally, the reason that the values of R or R2 do not depend on whether y is plotted against x, or vice versa, is evident in the definition provided by Eq. 1; specifically, there is no difference in how x and y are treated in that mathematical definition, so that x and y are interchangeable and mathematically symmetric. That is, if all of the x's were replaced by y's, and all the y's were replaced by x's, the equation would look the same. Thus, given any two column table of data, either column could be defined as x and the other as y, and the result returned by eq 1 would be the same.

REPLY TO COMMENT 2

The author agrees with the reviewer regarding the explanation on why values of R or R2 do not depend on whether y is plotted against x, or vice versa. The clarification made by the reviewer will be included in the revised manuscript. Any possible misinterpretations regarding the statement in line 45 and Eq. (1) will be removed from the revised manuscript.

COMMENT 3

Equation 4 and line 75ff. I have confirmed that Eq. 4 is correct, but the claim that the "deviations of X and Y from their means to obtain are assessed independently" is not. Specifically, the author has mistakenly concealed that dependence in his substitution for "m". The reader can refer back to Eq. 2 to see that m depends on the PRODUCT of the x and y deviations, and Eq. 4 depends on m; this conclusively refutes the author's statement.

REPLY TO COMMENT 3

The author agrees with the reviewer that m from Eq. (2) depends on the product of x and y deviations and Eq. (4) depends on m. In this case the statement that "….deviations of X and Y from their means to obtain gamma are assessed independently" can no longer hold water and will be removed from the revised manuscript. The point the author wanted to make was that R2 does not take into account model errors because it does not comprise squared differences between x and y as considered directly by other metrics Nash Sutcliffe Efficiency (NSE) and Root Mean Squared Error (RMSE).

COMMENT 4

I don't have time to plough through the author's derivation, but it is clear that computing this would be very difficult compared to many of the simple, single-formula metrics that are currently available to compare models and measurements.

REPLY TO COMMENT 4

The author agrees with the reviewer that the version of CMA in the discussion paper can be more computationally difficult than some existing metrics such as NSE and RMSE. Although the derivation of CMA appears long, the formula of CMA is as simple as other existing "goodness-of-fit" metrics. In summary, there are basically two formulae the author was presenting to address shortcomings of $R^2$. The first formula is measure of model efficiency (MME) and can be given by

MME = f $\times$ (min[mx, my]/max[mx, my]) $\times$(min[sx, sy]/max[sx, sy]) ................... .(1)

where f = absolute value of the coefficient of correlation between X and Y,

mx = squared mean of the x's,

my = squared mean of the y's,

sx = standard deviation of the x's,

sy = standard deviation of the y's,

min = minimum of two or more values, and

max = maximum of two or more values.

Computation of the MME is as simple as other existing "goodness-of-fit" metrics. The values of MME range from 0 to 1. MME=0 means the model indicates that the model is not better than the comparison baseline (or mean of observed series). MME=1 indicates that there are no model errors. In Eq. (1), MME takes into consideration co-variation of X and Y, difference in variances of X and Y, and how biased the mean of Y is from the mean of Xans of X and Y. However, a typical shortcoming of R2 which MME does not address it that when MME yields the same value in the two cases including (i) regressing Y on X, and (ii) when X is regressed on Y.

Based on the need to derive a metric (hereinafter referred to as Coefficient of Model Accuracy or Agreement CMA) such that it offers an all-encompassing solution to the shortcomings of $R^2$, we modify the term expressing ratio of standard deviations in terms of the sum of squared deviations of x's and h's from a common cb baseline herein taken as cb=3×(mx)^0.5. In other words, CMA=0 if mx=my=0, otherwise,

CMA= f × (min[mx, my]/max[mx, my]) × w1/w2 ....................(2)

where w1= sum of the minimum values of (x(i)-cb)^2 and (y(i)-cb)^2,

w2= sum of the maximum values of (x(i)-cb)^2 and (y(i)-cb)^2

Computation of CMA is as simple as other existing "goodness-of-fit" metrics. Given that the formula for correlation coefficient is already in-built in many of the computing software packages; it is very easy to automate MME and CMA. To allow applications of the new metrics, MATLAB and R codes as well as an illustrative MS Excel file to compute the MME and CMA will be provided for readers (in the form of supplementary materials to the revised manuscript).

COMMENT 5

I am unimpressed with the choice of the dataset used to illustrate application of the model. Data that are widely and readily accessible would be better, for example, data from a government website would be better. That original data set could be manipulated in various simple ways to compare the manipulation, now representing the "model", with the real data, and different metrics applied and compared.

REPLY TO COMMENT 5

Datasets used to compare the various "goodness-of-fit" metrics as presented in the discussion paper were selected over a catchment from a data scarce region. Indeed, hydro-meteorological datasets from such a region tend to generally have quality issues and possible attempts to deal with the data limitation problems by in-filling of missing data values can lower the accuracy of model predictions. In this line, the author agrees with the reviewer on the need to select and use another dataset which can easily be accessed by readers.

To take the reviewer's comment into consideration, hydro-meteorological datasets including daily catchment runoff, catchment-wide rainfall and evapotranspiration over Jardine River catchment in North Queensland, Australia were obtained from the website of "eWater toolkit" via https://toolkit.ewater.org.au/ (September 9, 2020). The streamflow data was for gauging station no. 927001 and the catchment area was 2500 km^2. The datasets can be found in a folder named "Data" under Rainfall Runoff Library (RRL) which can be downloaded upon online registration.

Outputs of hydrological models used for comparison of the various "goodness-of-fit" metrics were generated using automatic calibration strategy in terms of the Generalized Likelihood Uncertainty Estimation framework GLUE (Beven and Binley 1992). In this calibration strategy, GLUE technique is a Bayesian approach in which several parameters' sets are randomized from the prior distribution to infer the output (posterior) distribution based on the simulations. In short, there was no issue subjectivity in

obtaining model outputs for comparison of "goodness-of-fit" metrics.

Beven, K. J., and Binley, A. M.: The future role of distributed models: Model calibration and predictive uncertainty, Hydrol. Process., 6, 279–298, doi:10.1002/hyp.3360060305, 1992.

COMMENT 6

The extended narrative of the performance of different metrics is ineffective. A table whose first column lists the various properties, with additional columns to the right providing values for the particular metric that heads that column, would be far better and would allow direct and simple comparisons of the properties of each metric with all others. Is the metric constrained to range from 0 to 1? Are effects of bias easily excluded? How many different formulae are needed to compute the metric? Ease of computation: if not a trivial calculation, are automated programs readily and widely available, as they are for R2? Does the metric have real physical significance? Is it widely used? Etc.

REPLY TO COMMENT 6

The author is grateful to the reviewer for the constructive set of comments. Eventually, a table as shown in Fig.1 and Fig. 2 of this document will be provided in the revised manuscript. From Fig. 1 and Fig.2, it can be seen that CMA was compared with other metrics with respect to more than ten properties.

COMMENT 7

My conclusion is that the author has more work to do. A shorter, compact paper would be more effective.

REPLY TO COMMENT 7

Based on the reviewer's comments, a lot of work will be done during revision of the manuscript such as

(i) presenting a more considered CMA expression into a single formula,

(ii) tabulating comparison of CMA and other existing metrics,

(iii) eliminating extended narrative of the performance of various "goodness-of-fit" metrics,

(iv) revising the choice of datasets,

(v) revising the entire abstract,

(vi) removing possible misinterpretations regarding the shortcomings of R2.

To ensure the manuscript is short and straightforward, if comparison of performance of various metrics is to be done, it will be provided as supplementary material to the revised manuscript.
* * *
REPLY TO COMMENT No.6

| SNo | Property | "Goodness-of-fit" metric | | | | | | | |
|---|---|---|---|---|---|---|---|---|---|
| | | $R^2$ | NSE | CMA | MME | IOA | TSS | RMSE | MAB |
| 1 | Range | $0 – 1$ | $-\infty – 1$ | $0 – 1$ | $0 – 1$ | $0 – 1$ | $0 – 1$ | $RMSE \geq 0$ | $-\infty – +\infty$ |
| 2 | Error quantification* | Relative error measure | Relative error measure | Relative error measure | Relative error measure | Absolute error measure | Relative error measure | Absolute error measure | Relative error measure |
| 3 | Computation difficulty | Low | Low | Moderate | Low | Low | Moderate | Low | Low |
| 4 | Number of extra columns in Ms Excel (apart from the two columns of where X and Y are put) that can be used to compute the metric. | No column is required if an in-built formula is used, otherwise, 3 columns required. | 2 | 2 | None | 2 | 1 | 1 | 1 |
| 5 | Are automated programs available for its computation? | Yes | Can easily be automated | MATLAB, & RStudio codes available along with this paper | Can easily be automated | Can easily be automated | Can easily be automated | Yes | Yes |
| 6 | Does the metric measure co-variation of observed and modeled series. | Yes | No | Yes | Yes | No | Yes | No | No |
| 7 | How many sub-formulae required to compute the metric | 1 | 1 | 2 | 2 | 1 | 2 | 1 | 1 |

* For an absolute error measure, the difference between observed and modeled data is obtained in terms of the unit of the observed variable. In relative error measure, the mismatch between observed and modeled data is quantitatively evaluated from zero to one. Here, values of zero and one indicate no relationship and perfect agreement, respectively (Legates and Davis, 1997).

**Fig. 1.** Comparison of CMA and other metrics (part 1)

| SNo | Property | "Goodness-of-fit" metric | | | | | | | |
|---|---|---|---|---|---|---|---|---|---|
| | | $R^2$ | NSE | CMA | MME | IOA | TSS | RMSE | MAB |
| 8 | Application or use of the metric | Widely used | Widely used | New | New | Widely used | Widely used | Widely used | Widely used |
| 9 | Sensitivity of squaring differences between observed and model output on large values | No squared differences between observed and modeled values | High | No squared differences between observed and modeled values | No squared differences | High | No squared differences between observed and modeled values | High | No squared differences between observed and modeled values |
| 10 | Increase towards to its maximum | Fast | Moderate | Moderate | Moderate | Fast | Fast | Moderate | Moderate |
| 11 | Physical relevance of the metric. Does it have unit? | No unit | No unit | No unit | No unit | No unit | No unit | Has unit of the observed variable | No unit |
| 12 | Metric's value in regressing X on Y compared to the case when Y is regressed on X | Same | Different | Different | Same | Same | Same | Same | Different |
| 13 | Sensitivity of the metric to possible outliers | High | High | Low | Low | High | High | High | High |

**Reference**

Legates, D. R. and Davis, R. E.: The continuing search for an anthropogenic climate change signal: limitations of correlation-based approaches, Geophys. Res. Lett., 24, 2319–2322, doi: 10.1029/97GL02207, 1997.

**Fig. 2.** Comparison of CMA and other metrics (part 2)

---

## Short Comment (SC2) · 3 Oct 2020

This paper explores how we go about measuring the "goodness of fit", proposing an alternative measure to $R^2$. The author cites Kvalseth's 1985 paper (Cautionary Note and $R^2$), but fails to acknowledge that there are more than 1 measures that are commonly used which are called $R^2$ (or the coefficient of determination). The most common of these are:

1. $R^2$ for a linear regression (square of the Pearson Product Moment Correlation Coefficient – note that for a sample of a population, this is traditionally called $r$). This is $R_6^2$ in Kvalseth (1985)

2. NSE (called $R^2$ in the Nash Sutcliffe paper – note that the Nash Sutcliffe paper built

from this existing definition of $R^2$ to propose a way of evaluating incremental changes in a model by replacing the mean observed flow in the denominator with the previous version of the modelled flow – which they called $r^2$ to discriminate this from what Hydrologists refer to NSE). This is $R_1^2$ in Kvalseth (1985).

3. Fraction of variance explained (1-variance of model residuals/variations of observed output). This is identical to NSE if the bias is zero. It is very close to NSE for any reasonable model (i.e. small bias), though this measure will always be larger than NSE (indicating a slightly better fit) if the bias is not zero. This is $R_4^2$ in Kvalseth (1985).

The fact that $R^2$ (or the coefficient of determination) is multiply defined leads to a lot of confusion, and authors need to be clear about which version they are using, and readers need to ensure that they understand which version is being referred to. While there is growing use of the $R^2$ for a linear regression in Hydrology (something that really needs to stop), many of the papers that refer to $R^2$ or the coefficient of determination are actually using NSE. My view is that the linear regression version adds little to NSE, and if the linear regression form is used, the slope and intercept of the regression should always be reported as well.

The author also does not report on the conclusion of the analysis done by Kvalseth – that generally the best form of $R^2$ appears to be what hydrologists refer to as the NSE.

The author refers to example hydrology papers that have used $R^2$. Looking at some of these papers, they appear to use the NSE version of $R^2$. However, the author also refers to the regression-based version of $R^2$, specifically in the statement: "Regressing X on Y yields $R^2$ which is the same as that if Y is regressed on X thereby invalidating its use as a coefficient of determination" in the abstract and body of the paper. These two forms of $R^2$ are not equivalent (though related in the unbiased case – see Bardsley, 2013, Hydrol. Proc. DOI: 10.1002/hyp.9914), and the author needs to clarify which $R^2$ is being discussed (i.e. focus on NSE) and remove the discussion on the linear regression. Note that for NSE, the value doesn't remain the same if you swap the

observed and modelled values, so the criticism of the linear regression form is irrelevant in regard to the NSE form.

Overall, this confusion about which $R^2$ is being talked about distracts the reader from the message the author is trying to convey. Subsequently the introduction requires extensive revision. The author would be better served by not focusing on the $R^2$ for a linear regression (other than to state that this should not be used), and focus on the NSE form of $R^2$. This would substantially change the discussion part of the paper also (as well as impacting on the abstract and conclusion).

Given the issue with the discussion of $R^2$ in the paper, there is merit in the coefficient of model accuracy being proposed by the author. Each performance criteria gives a different view of the model behavior. Some (e.g. NSE and RMSE) are very similar (in the case based on the sum of squared residuals), so little is gained from using both of these. The CMA proposed by the author is different from any performance criteria I have seen, being more closely related to the Spearman correlation coefficient (noting the penalty based on comparison of individual observed and modelled values), and therefore may be of benefit for modellers.

I spotted a few typos in the paper, but didn't check specifically for these. The author should carefully check the paper for errors.

**Specific comments**

1. Line 64: I find the use of the upper case to indicate the series of values and the lower case with subscript I to represent individual values in the series an unnecessary complication. Adding the subscript indicates that you are looking at individual values, so the data set could be lower case without confusion. If you want to use the upper case for the series, then logically, the mean values in the expressions should also be upper case as these indicate the mean value of the series). At the moment, this leads to a confusing mixture of notation in the paper (e.g. line 88, which refers to the dataset Y, and gives values of y=...). It would make more sense to refer to one as the original

(untransformed values), and the other as the transformed values (e.g. x=X-mean(X); y=Y-mean(X)).

2. Line 105-109: Presumably, the author is using x=observed-mean observed, and y=modelled-mean observed?

3. Line 113: Maybe I am missing something, but with the subscript i includes in equations 9 and 10, isn't $\min(h_i,x_i)=\max(h_i,x_i)$ as $h_i$ and $x_i$ are scalars? This is pivotal for calculating the $\beta$, and so needs to be clarified. Also in equation 10, I assume this should be $\omega_{2,i}$?

4. Line 116: Why not use the absolute value rather than the squared value to handle the issue of opposing signs? Is there any reason for the choice, or is it just a personal preference? What impact does this have on the result?

5. Line 152: MAB is defined as the model average bias. However, the inclusion of the denominator means this is not an average bias. Please check the formula given in the paper. It should also be noted that this formulation is problematic if the observed values are too close to zero. Is this based on a published formation? If so, please add a citation.

6. Line 172: either "poorly fitted model" or "poor model fit"

7. Line 176: "tend to zero"

8. 190: Yes, $R^2$ for a linear regression can be high for poor models, if the model error can be well modelled by a linear relationship with the observed values. This is why the $R^2$ for a linear regression should never be used (unless you also report the slope and intercept). Again, I point out that many hydrological papers that refer to $R^2$ or the coefficient of determination are using NSE rather than the $R^2$ for a linear regression.

---

## Author Comment (AC3) · 24 Oct 2020

GENERAL

The author is thankful to Barry Croke for his meticulous comments which were all constructive in enhancing both quality and content of the paper. Furthermore, the author is grateful for the recognition that there is merit in the coefficient of model accuracy (CMA) being proposed.

MAJOR COMMENTS

The author agrees with Barry that there are various versions of R-squared. Kvalseth (1985) in his paper (The American Statistician, 39:4, 279-285, DOI: 10.1080/00031305.1985.10479448) elaborated on nine versions of R-squared (see

[Figure]

Eqs (1)-(9) and (11) of cited paper). It is important that clarifications are required on the existing variants of R-squared and these will be made in the revised manuscript.

In Geoscience (take for instance, the field of hydrology), the versions of R-squared which are increasingly being used are (1) the square of the Pearson product moment correlation coefficient, and (2) coefficient of determination or the Nash-Sutcliffe Efficiency NSE (Nash and Sutcliffe 1970, J. Hydrol., 10, 282–290, doi:10.1016/0022-1694(70)90255-6). These two R-squared versions (1) and (2) as correctly noted by Barry are not the same based on the information from Bardsley (2013) published in Hydrol. Proc. DOI: 10.1002/hyp.9914. The author also realizes that clarification, for instance, on the two common versions of R-squared was lacking in the discussion paper and this will be worked on during revision of the manuscript. Furthermore, the author agrees that information in the manuscript is somewhat mixed up regarding which version is being treated. Although the version of R-squared based on linear regression is rampantly used, the revised manuscript will focus on the NSE as suggested. This follows the conclusion of Kvalseth (1985) that generally the best form of R-squared appears to be what hydrologists refer to as the NSE. To make the need of focusing on NSE emphatic, the title will slightly be changed from " From R-squared to coefficient of model accuracy for assessing "goodness-of-fits"" to " From R-squared or Nash Sutcliffe Efficiency to coefficient of model accuracy for assessing "goodness-of-fits"". Furthermore, discussion regarding the version of R-squared based on linear regression will be removed from the revised manuscript. The criticism that R-squared still remains the same when X and Y are swapped will also be removed from the revised manuscript. Generally, due to the need to put NSE central to the manuscript, several sentences in the abstract, conclusion, discussion, and other sections will be removed or revisited.

Results of comparison of NSE, CMA, Taylor Skill Score TSS (Taylor (2001), J. Geophys. Res., 106, 7183−7192, DOI: 10.1029/2000JD900719), and the version of R-squared recommended by Kvalseth (1985) (or Eq. 11 of the paper by Kvalseth (1985)) can be seen in Figures 1 and 2 of this reply to Barry Croke's comments. The need to consider

TSS in comparison CMA with existing metrics was recommended by Ulrich Schumann in his short comment on this discussion paper (see Geosci. Model Dev. Discuss., DOI: 10.5194/gmd-2020-51-SC1). Root Mean Squared Error (RMSE) was not considered for comparison because it is deemed comparable to NSE since both of them are based on the sum of squared residuals. Figure 1 shows comparison of values of the various "goodness-of-fit" metrics NSE, CMA, TSS, and IoA. Figure 2 comprises comparison of observed and modeled flow based on calibration of two models including the Hydrological Model focusing on Sub-flows' Variation (HMSV) and Nedbør-Afstrømnings-Model (NAM). Generally, the results show the acceptability of the CMA being introduced. The reason behind the performance of CMA can be explain in terms of a careful derivation of the proposed CMA's formula. Based on comments from other reviewer's, CMA's formula from the discussion paper requires some modification and this will be done in the revised manuscript as described next.

For CMA to offer an all-encompassing solution to the shortcomings of R-squared, it has three components including (i) absolute value of rank-based coefficient of correlation (f) between observed (X) and modeled (Y) series, (ii) difference in the variances of X and Y, and (iii) bias of the mean of Y from the mean of X.

The first CMA part comprising the term f quantifies co-variability of X with Y. Values of f ranges from zero to 1. When X and Y are perfectly correlated, f takes the value of one. When there is no correlation between X and Y, f yields a value of zero.

In the second part, we obtain another term Wg in terms of the amounts by which X and Y vary from a common baseline (Cb). This Cb is taken to be three times the mean of observed series (Xm); in other words, Cb=3Xm. In Cb, Xm is multiplied by 3 not 2 or 1 due to the need to maintain a reasonableness of CMA. Values of Cb greater (less) than 3 makes CMA conservative (exaggerated). Finally, Wg is given by the ratio of w1 to w2 where where w1= sum of the minimum values of (X-Cb)ˆ2 and (Y-Cb)ˆ2, and w2= sum of the maximum values of (X-Cb)ˆ2 and (Y-Cb)ˆ2. When deviations of the values of X from Cb are the same as those for Y, Wg yields a value of one. It is important to note

that, here, Wg can have a value of one regardless of how far apart Xm is from mean of Y (or Ym). Furthermore, Wg can have a value regardless of whether the X and Y are correlated or not. This means, CMA should take into account how biased Ym is from Xm. This leads us to the third term of CMA denoted by Vm.

The third part of CMA is such that Vm = 1 if Xm = Ym otherwise Vm is the ratio of of u1 to u2 where u1 is the squared minimum value among Xm and Ym, and u2 is the squared maximum value among Xm and Ym.

In summary,

CMA= f × Vm ×Wg . . .. . .. . .. . .. . .. . .. . .. . .. . .. . .. . .. . .. .(1)

CMA ranges from 0 to 1. When CMA is zero, it indicates that the model is not better than the comparison baseline (or mean of observed series). CMA value of one indicates that there are no model errors. In the case when CMA is one, it means (i) X and Y are perfectly correlated, (ii) the deviations of X and Y from Cb are the same, and (iii) Xm is equal to Ym.

Mathematical expressions involved in the derivation of CMA will be clearly presented in the revised manuscript. The reviewer noted that there are a few typos in the paper. The author agrees with the reviewer. Corrections will be made in the revised manuscript.

MINOR COMMENTS

1. Line 64: The reviewer noted that the use of both upper and lower letters in representing series and its individual values is confusing. The author agrees with the reviewer on this. One style of representation or notation for series and its individual values will consistently be adopted as advised by the reviewer.

2. Line 105-109: Regarding penalty, the reviewer wondered if we are making use of x=observed(x)-mean of observed and y=modeled-mean of observed (x). However, we are making use of observed (x) and modeled (y).

3. Line 113: Regarding sub-scripts in equations 9 and 10, we are determining the deviation of x and h from the comparison baseline (Cb). For instance, min(xi,hi) means whichever of the ith values of x and h is less than the other. On the other hand, max(xi,hi) means whichever of the ith values of x and h is greater than the other. So, max(xi,hi) is not the same as max(xi, hi). These components in the revised equation for CMA are in terms of w1 and w2 as explained shortly before. The number of terms in the revised CMA formula were reduced for clarity.

4. Line 116: Since we are characterizing variance, we make use of squared deviations. Furthermore, effect of considering absolute and squared values can influence CMA. For instance if we have a=0.2 and b=0.4, it means a^2=0.04, b^2=0.16. Dividing these two terms a and b we obtain a/b=0.5 and a^2/b^2=0.25. In other words, the use of square values instead of absolute values is to balance the contribution of the term Wg (see Eq. 1) in CMA.

5. Line 152: The author agrees that including the observed value as a denominator in the formula of MAB (taken to mean model average bias) can be problematic. In the revised manuscript, MAB will be left out so as to allow comparison of "goodness-of-fits" which do not have units.

6. In the revised manuscript, "poorly fit model" will be changed to "poorly fitted model".

7. Line 176: In the revised manuscript "tend o zero" will be changed to "tend to zero".

8. Line 190: The author agrees with the reviewer on the fact that hydrological papers that refer to R-squared or the coefficient of determination are using NSE rather than the R-squared for a linear regression. For clarity in the revised manuscript, focus will be given to NSE instead of R-squared based on linear regression.
* * *
[Figure]

**Fig. 1.** Comparison of various "goodness-of-fits" including a)-b) CMA, IoA, TSS, NSE

**Fig. 2.** Comparison of observed and modeled flow from a)-d) HMSV and e)-h) NAM